# SUBJECTIVE REINFORCEMENT LEARNING FOR OPEN COMPLEX ENVIRONMENTS

## ABSTRACT

Solving tasks in open environments has been one of the long-time pursuits of reinforcement learning research. We propose that *data confusion* is the core underlying problem. Although there exist methods that implicitly alleviate it from different perspectives, we argue that their solutions are based on task-specific prior knowledge that is constrained to certain kinds of tasks and lacks theoretical guarantees. In this paper, *Subjective Reinforcement Learning Framework* is proposed to state the problem from a broader and systematic view, and *subjective policy* is proposed to represent existing related algorithms in general. Theoretical analysis is given about the conditions for the superiority of a subjective policy, and the relationship between model complexity and the overall performance. Results are further applied as guidance for algorithm design without task-specific prior knowledge.

## 1 INTRODUCTION

One important target of reinforcement learning (RL) is to realize an autonomous agent that can help people solve various kinds of tasks in open environments, where the agent's observations and the environment's dynamics is both diverse and unpredictable (Wu et al., 2019). For example, a vision-based robot doing housework in all kinds of room-settings may observe various combinations of indoor scenes; different tasks may require it to take different actions under the same observation; the successor observation after performing an action is not predictable, because the shape of rooms, furniture settings and pets' movements, etc. are not known a priori. Classical RL models environment and tasks as a Markov Decision Process (MDP). From this perspective, tasks in open environments pose difficulties that the state space $\mathcal{S}$ is large, the state transition dynamics $P$ is nonstationary, and the reward function $R$ may be contradictory(Brodeur et al., 2017; Xu et al., 2018).

We argue that the key problem underlying these difficulties is *data confusion*, that is every sample the agent acquires from an open environment may reflect the properties of only a small part of the entire environment for a specific time period (Kempka et al., 2016), thus data samples with similar current observations may have contradictory meanings under different circumstances, i.e. different transition dynamics, state (or state-action) values or action preferences. For this problem, current RL methods introduce out-of-MDP information such as task-encoding (Zhu et al., 2016) and goal-description (Wang et al., 2018), or expand the state space into history space, in which every element is the concatenation of all information gained so far $\tau_t = s_{1:t}$ (Farias et al., 2007). As for the former category, the quality and quantity of extra information are decided by the designer (Schaul et al., 2015). As for the latter category, although the whole history is what we can use best to alleviate data confusion in theory, it increases the complexity to a maximum point and is hard to achieve in practice; algorithms in this category thus either introduce regularization terms or transform the optimization target to easier ones. The problem is that both these categories depend on humans' understandings to certain tasks and lack theoretical guarantees; consider that tasks in open environments are diverse and unpredictable, it is necessary to build a system that can dynamically fit tasks without prior knowledge.

Intuitively, we humans also face data confusion problems in the real world, e.g. when going to office from home, different modes of transportation may include conflict behaviors e.g. walking in opposite directions. However, as long as we utilize arbitrary information and subjectively choose one of the modes, e.g. taking the bus, this task becomes concrete and without confusion; also we may

subjectively focus on more concise aspects of the problem e.g. selecting bus numbers, which make the tasks simpler. Inspired by this, we propose a novel framework named *Subjective Reinforcement Learning Framework* that models the environment and tasks as a set of simple MDPs with no data confusion problem, and a subjectivity model over them; the corresponding *subjective policy* then includes two parts: acquiring subjectivity for each state with *subjective samples*, and choosing an action under the selected subjectivity. From this perspective, all aforementioned existing methods can be summarized as introducing subjective samples, i.e. state history or extra task information, beyond current observation $s_t$ to form a subjectivity so that the resulting subjectivity-conditioned policy does not suffer from data confusion problem and thus have better performance. Therefore they can be recognized as instantiations of our subjective policy, and analysis based on our framework is general and meaningful for all these methods.

In this paper, we discuss the conditions under which the subjective policy can achieve better results, and analyze in theory the relationship between overall performance and some key elements that characterize a subjective policy. The key elements are concrete and modifiable, and the analytical results guide us to design policy models that fit the need of problems in open complex environments without task-specific prior knowledge and with theoretically-guaranteed performance.

To sum up, we think our contributions are as follows:

1. To our best knowledge, we are the first to point out data confusion as a key problem to solving tasks in open complex environments;

2. We propose a novel framework named Subjective RL Framework and give theoretical analysis from a broader and systematic view;

3. We provide guidances for algorithm design based on our theoretical results.

## 2 RELATED WORK

Our subjective RL framework deals with data confusion problem facing tasks in open environments, which has also been implicitly considered in several RL fields: multi-task RL (Zhu et al., 2016), meta-RL (Rakelly et al., 2019), continuous adaptation (Yu et al., 2018) and hierarchical RL (Al-Emran, 2015). We analyze their inherent defects—dependence on designers' knowledge for tasks—in the following three paragraphs.

The field of multi-task RL considers an agent performing different tasks in possibly different environments, where data confusion is expected to happen across tasks. One direct category of methods equips the agent with extra task-specific information (Riedmiller et al., 2018), e.g. a vector $g_i$ describing the goal of task $T_i$; these methods requires the extra information to be available and sufficient enough to distinguish tasks from each other, which is guaranteed by designers' knowledge. Considering cases where suitable extra information is not available, various task-encoding methods have been proposed (Sung et al., 2017), which generally extract a task feature vector from the whole trajectory $\tau$ and build the policy under condition of such feature vectors. These methods can be concluded as adding regularization terms into the complex end-to-end framework according to experts' prior knowledge, which includes the length (Sorokin & Burtsev, 2019) and prior distribution (Rakelly et al., 2019) of the feature vector.

Meta-RL methods, or learning-to-learn, turns to model the learning process instead of the original task (Gupta et al., 2018a;b), which implicitly bypass the problem of data confusion. However, these methods still treat the learning process as a single-value function, e.g. the learning process of task embedding (Lan et al., 2019) and an exploration strategy (Garcia & Thomas, 2019); thus their successes are based on the assumption that there are no data confusion in the meta learning phase. Similarly, continuous adaptation problems focuses on the transfer performance from source to target tasks and neglect the data confusion problem by assuming that the relationship between successive tasks can be correctly modeled by single-value mappings (Al-Shedivat et al., 2018).

Hierarchical RL is a broad field where methods aim to divide complex tasks into many different sub-tasks so that the overall data-efficiency can be improved (Barto & Mahadevan, 2003; Al-Emran, 2015). Although this idea of divide-and-conquer may result in the confusing data samples being divided into different sub-models and thus each sub-model handles no confusion, which is one typical target of subjectivity in our framework, the actual performance is determined by the design

of the hierarchical structure, e.g. number of options (Jain et al., 2018) and capacity of both the top-level and low-level models, which are still designed according to prior knowledge and adjusted through empirical results. Instead, our framework takes all these hyper-parameters in consideration and can adjust to tasks without prior knowledge.

The data confusion problem in open environments can also be recognized from the perspective of errors in function approximation. Existing research on function approximators in RL consider some specific types of model like linear approximations or neural networks, and analysis are given about the overall convergence or performance of certain RL algorithms (Schoknecht, 2003; Achiam et al., 2019; Papavassiliou & Russell, 1999). However, approximators considered are end-to-end and do not consider the utilization of extra information. Different from these, our framework formalizes both original data samples and extra information used to construct subjectives, and so has the capacity to analyze the effect of extra information.

## 3 SUBJECTIVE REINFORCEMENT LEARNING FRAMEWORK

In this section we describe the data confusion problem of tasks in open environments. Then we give mathematical formulation of our subjectivity reinforcement learning framework that states the data confusion problem from a broader and systematic view.

### 3.1 PROBLEM STATEMENT

In traditional RL, the environment and tasks are modeled as $\langle T, \mathcal{S}, \mathcal{A}, P, R, \gamma, \rho \rangle$, where $T = \{0, 1, 2, \dots\}$ is the set of considered time steps, $\mathcal{S} = \{s_t\}$ the state space, $\mathcal{A} = \{a_t\}$ the action space, $P(s_{t+1} | s_t, a_t)$ the transition probabilities from each state-action pair to successive states, $R : \mathcal{S} \times \mathbb{R} \to [0, 1]$ the reward distribution over states that expresses the task, $\gamma$ the discount factor that describes the relative importance between short-sight and far-sight rewards, and $\rho$ the probabilistic distribution of initial observations. The goal of traditional RL is to find a policy $\pi_T : \mathcal{S} \to \mathcal{A}$ that maximizes the following return:

$$\max_{\pi_T \in \Pi_T} G(\pi_T) = \sum_{s_0 \in \mathcal{S}} \rho(s_0) \left\{ \sum_{t=1}^{\infty} \gamma^t \int_r [r R(r | s_t) \, \mathrm{d}r] \right\} = \sum_{s_0 \in \mathcal{S}} \rho(s_0) V_{\pi_T}(s_0) \tag{1}$$

where $V_{\pi_T}(s)$ is the state-value function that satisfies the following Bellman equation:

$$V_{\pi_T}(s) = \sum_{a \in \mathcal{A}} \pi_T(a|s) \sum_{s' \in \mathcal{S}} P(s' | s, a) \int_r R(r | s') [r + \gamma V_{\pi_T}(s')] \, \mathrm{d}r \tag{2}$$

RL agents interact with the environment to collect transitions $\langle s_t, a_t, s_{t+1}, r_{t+1} \rangle$. Different RL algorithms utilize the collected transitions and equation (2) in different ways to construct functions that represent polices so as to maximize objective (1). Generally they can be summarized into three categories: model-constructing, value-based, and direct action-mapping, which respectively process the transitions into forms of $(s_t, a_t) \to s_{t+1}$, $s_t \to V$(or $(s_t, a_t) \to Q$ for state-action value-based methods) and $s_t \to a_t$. In this paper we call such correspondences "data samples" to emphasize that they are the actual data used to construct function that represent the policy, and use $z = (x, y)$ to generally represent the content and label of a data sample. Further, we denote the function as $g_T(\cdot)$, and the optimization problem becomes:

$$\min_{g_T} \sum_{i=1}^{N_D} \mathcal{L}[y_i, g_T(x_i)] \tag{3}$$

where D is the set of data samples and $N_D$ is the number of them; $\mathcal{L}$ is a predefined loss function, e.g. l2-norm.

Note that data samples reflect properties of environment, policy and tasks, and they depend on the sampling process in RL. Because $g_T(\cdot)$ is a single-value mapping, normally the optimization problem (3) results in the expectation of data samples and alleviate the variances induced by the sampling processes. However, in open environments the observation space $\mathcal{S}$ is large, the transition dynamics $P$ is nonstationary, and reward function $R$ may be contradictory; thus same state-action

pairs may transit to contradictory successive states and rewards; then the Bellman equation (2) may return different label $y$ for the same data content $x$. Such differences come from the variances in environment or tasks, and should not be entangled through expectation. In this paper we name this problem *data confusion*. To eliminate this problem, extra information, which we denote as $\kappa$, need to be introduced, and the key problem is how to select $\kappa$ and efficiently use it to eliminate data confusion at the minimum cost of simplicity. In the next subsection we introduce Subjective RL Framework to formalize this problem.

As aforementioned, methods that expand the state space directly into the history space and try to construct the mapping from history to labels does not necessarily eliminates this problem in practice, so it remains meaningful to analyze the data confusion problem under the assumption that function approximators can only handle finite length of states. To make notations simple, here we focus on confusing mapping from single states and leave the history to the subjectivity part, without loss of generality.

## 3.2 FRAMEWORK

We propose to model tasks in open environments as $\langle \boldsymbol{h}, M^k \rangle$, where $\boldsymbol{h}$ is the *subjectivity* that "subjectively" treats the entire large, nonstationary and possibly contradictory environment as some separate, simple and stationary MDPs $M^k = \langle T^k, \mathcal{S}^k, \mathcal{A}^k, P^k, R^k, \gamma^k, \rho^k \rangle$, which we call *subjective MDPs*. One key property of subjective MDP is that there are not data confusion problems for the functions used to construct corresponding policies, i.e $\pi_{\mathrm{S}}^k (a|s)$, $V_{\mathrm{S}}^k (s)$, $Q_{\mathrm{S}}^k (s,a)$ and $P_{\mathrm{S}}^k (s,a)$ should not contain data confusions. The expected role of $\boldsymbol{h}$ is to utilize extra information $\kappa$ to divide contradictory data samples into different sets. A formal definition of it is given below.

**Definition 1.** The *subjectivity* $\boldsymbol{h}$ in subjective RL framework is a function that maps a piece of extra information $\kappa$ to the sum-to-one vector that corresponds to the weights of all possible subjective MDPs $M^k$.

Correspondingly, we maintain a *subjective policy* $\pi_{\mathrm{S}}^k (a|s, \kappa)$ for each subjective MDP and the overall policy can be formulated as:

$$\pi_z (a|s, \kappa) = \sum_{k=1}^{N_{\mathrm{S}}} h_k (s, \kappa) \pi_{\mathrm{S}}^k (a|s) \qquad (4)$$

where $h_k$ is the $k$-th element of $\boldsymbol{h}$ and $N_{\mathrm{S}}$ is a variable characterizing the number of maintained subjective MDPs; $\kappa \sim F(\kappa)$ where $F(\kappa)$ is a distribution that is assumed to be available. Here $\kappa$ may correspond to multiple types of information, including state history, out-of-MDP task encoding, samples from related tasks, etc. Similarly the overall value functions or model transitions can be represented in similar forms, e.g. $V_{\pi_z} (s, \kappa) = \sum_{k=1}^{N_{\mathrm{S}}} h_k (s, \kappa) V_{\mathrm{S}}^k (s)$.

Then the overall global optimization problem of the task becomes:

$$\max_{\pi_z \in \Pi_z} G (\pi_z) = \sum_{s_0 \in \mathcal{S}} \rho (s_0) \left\{ \sum_{t=1}^{\infty} \gamma^t \int_r [r R (r | s_t) \, \mathrm{d}r] \right\} = \sum_{s_0 \in \mathcal{S}} \rho (s_0) V_{\pi_z} (s_0) \qquad (5)$$

where

$$V_{\pi_z} (s, \kappa) = \sum_{a \in \mathcal{A}} [\pi_z (a|s, \kappa)] \sum_{s' \in \mathcal{S}} \left\{ \sum_{k=1}^{N_{\mathrm{S}}} h_k (s, \kappa) P^k (s' | s, a) ] \right.$$
$$\left. \cdot \sum_{k=1}^{N_{\mathrm{S}}} h_k (s, \kappa) \int_r \left\{ R^k (r | s') [r + \gamma V_{\pi_z} (s', \kappa)] \, \mathrm{d}r \right\} \right\} \qquad (6)$$

Note that the subjectivities only disentangle the original problem into a variable number of subjective MDPs and does not change the original properties, thus the global return $G$ reflects the same optimization goal.

From the perspective of data samples, the policy function $g_z(\cdot)$ becomes:

$$g_z (x, \kappa) = \sum_{k=1}^{N_{\mathrm{S}}} h_k (x, \kappa) g_{\mathrm{S}}^k (x) = \boldsymbol{h} (x, \kappa) \cdot \boldsymbol{g}_{\mathrm{S}} (x) \qquad (7)$$

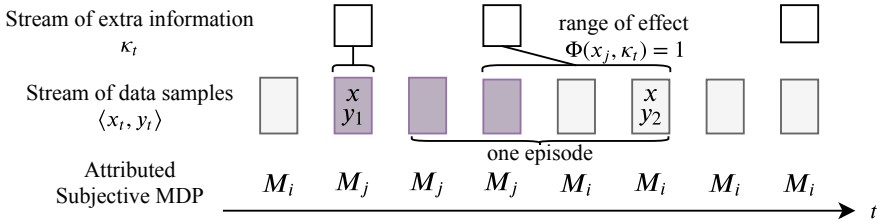

Figure 1: Samples in the Subjective RL Framework

where $g_S^k$ is the $k$-th mapping for data samples from the corresponding subjective MDP, and $\boldsymbol{g}_S$ is the concatenation of all the mappings.

One important property of extra information $\kappa$ is that it may become available only after certain time step and its effect on subjectivity about data samples may only last for a finite time-period, e.g. the history of states $\tau_t$ appear at time-step $t$ and expires after the possibly existing episode ends. Therefore we use $\kappa_t$ to denote the extra information that becomes available at time-step $t$, and introduce a data-related mask to characterize its range of effect: $\Phi\left(x_j, \kappa_t\right) = 1$ if $\kappa_t$ can affect the subjectivity about data content $x_j$, and 0 otherwise. Note that $\Phi$ only depends on the arriving time of extra information and the data samples, so it should be available under most conditions. Then the constraint on $\boldsymbol{h}$ can be formalized as

$$h_k\left(x, \kappa_t\right) = \frac{1}{N_S} \ \ \text{if} \ \ \Phi\left(x, \kappa_t\right) = 0 \ \ \text{for} \ \ k = 1, 2, \dots, N_S \tag{8}$$

Figure 1 provides an illustration of the relationship between data samples, extra information and the mask.

We parametrize $\boldsymbol{h}$ and $\boldsymbol{g}_S$ with $\alpha_h$ and $\alpha_S$ respectively, and require the range $\alpha_h$ to satisfy the constraint expressed in eq.(8). The optimization problem about data samples then becomes:

$$\min_{\alpha_h, \alpha_S} \frac{1}{N_\kappa} \sum_{j=1}^{N_\kappa} \frac{1}{N_D} \sum_{i=1}^{N_D} \mathcal{L}\left[y_i, \boldsymbol{h}\left(x_i, \kappa_j\right) \cdot \boldsymbol{g}_S\left(x_i\right)\right] \} \tag{9}$$

where $N_\kappa$ is the number of available extra information.

## 4 THEORETICAL ANALYSIS

In the above section we formalize our Subjective Reinforcement Learning Framework which formally considers the data confusion problem and the extra information used to eliminate it. Below we analyze in theory the performance of a subjective policy. In the first part we focus on the return $G$ which is the optimization target of RL; in the second part we turn to analyze the risk bound of the policy function which relates closely to $G$, is general, and does not depend on certain algorithms.

### 4.1 RETURN GAP

In reinforcement learning, the return $G$ is the overall criterion to evaluate the performance of a policy. Here we firstly define the return gap as:

$$\delta = \max_{\pi_z \in \Pi_z} G\left(\pi_z\right) - \max_{\pi_T \in \Pi_T} G\left(\pi_T\right) \tag{10}$$

where $\pi_z$ refers the subjective policy and $\pi_T$ the policies in traditional forms.

For tasks in open environments, there may be more than one optimal trajectories $\tau^*$ that maximize the global return. For simplicity in notation here we consider cases where there is only one initial state $s_0$ and only one optimal trajectory, without loss of generality. Intuitively the loss function (3) and (9) need not be zero for an optimal policy, but the policy function must represent data samples from optimal trajectories well so as to be optimal; thus when data confusion problem happens within such data samples, the subjective policy may gain better performance.

In the following analysis we denote the optimal subjective policy as $\pi_z^*$ (which includes $h_k^*$ and $\pi_{\mathrm{S}}^{k*}$) and let the optimal trajectory achieved by it be $\tau_z^* = \{(s_0, a_0), (s_1, a_1), \ldots, (s_T, a_T)\}$. In order to express the changing of extra information $\kappa$, we use subscript i.e. $\kappa_t$ to denote the available extra information at time $t$.

**Theorem 1.** *Assume that all possible data samples appear infinite times when the total number of samples $N_{\mathrm{D}} \to \infty$, then the return gap $\delta \geq 0$ holds true;*

*Proof.* In our framework we have subjectivity $\boldsymbol{h}$ that aims to divide confused data samples under the constraint of eq.(8). Consider a given piece of extra information $\kappa$ and any subjectivity $\boldsymbol{h}_0$, our subjective policy is defined as eq.(4). As for any traditional policy $\pi_{\mathrm{T}}(a|s)$, we can define fake subjective policies as:

$$\pi_{z,\mathrm{fake}}(a|s, \kappa) = \sum_{k=1}^{N_{\mathrm{S}}} h_{0,k}(s, \kappa)\, \pi_{\mathrm{S},\mathrm{fake}}^k(a|s) \tag{11}$$

where $\pi_{\mathrm{S},\mathrm{fake}}^k(a|s) = \pi_{\mathrm{T}}(a|s)$ for any $s, a, k$. Then we have $\pi_{z,\mathrm{fake}}(a|s) = \pi_{\mathrm{T}}(a|s)$ hold true for any $s, a, \kappa$, which means the policy space $\Pi_{\mathrm{T}}$ is equally transformed in to $\Pi_{z,\mathrm{fake}}$. Because $\pi_{\mathrm{S},\mathrm{fake}}^k(a|s)$ are defined to be the same with reference to $k$, we have $\Pi_{z,\mathrm{fake}} \subseteq \Pi_z$. Therefore, under the condition that samples required to estimate the optimal policy are plenty, which ensures that an optimal policy can be learned out of the data, the introduced subjectivity extends the policy space and theoretically provides no worse result than single-value-based methods, that is $\delta \geq 0$. $\square$

**Corollary 1.** *Further, if there exists steps $i, j \in \{1, \ldots, T\}$ such that transitions from $\tau_z^*$ satisfies $s_i = s_j$ but $\pi_z(a|s_i, \kappa_i) \neq \pi_z(a|s_j, \kappa_j)$, then $\delta > 0$.*

*Proof:* We have $\pi_{\mathrm{T}}(a|s_i) = \pi_{\mathrm{T}}(a|s_j)$ because $s_i = s_j$. Because of $\pi_z(a|s_i, \kappa_i) \neq \pi_z(a|s_j, \kappa_j)$, $\pi_{\mathrm{T}}(a|s_i) = a_i$ and $\pi_{\mathrm{T}}(a|s_j) = a_j$ cannot hold at the same time. Thus $\tau_z^*$ cannot be fully presented by $\pi_{\mathrm{T}}$. As analyzed in part (1), the policies that can be represented by $\pi_{\mathrm{T}}$ is a subset of $\pi_z$, and according to the assumption that $\tau_z^*$ is unique, we reach the conclusion that $\delta > 0$. $\square$

The following theorem transforms the return gap into the analysis of risk bound of the policy functions:

**Theorem 2.** *Under the assumption of sufficient sampling and exploration, the gap between approximate return and the optimal return satisfies the following inequality:*

$$\lim_{w \to \infty} |G(\pi_w) - G^*| \leq \frac{\delta_{\mathrm{com}} + 2\gamma\epsilon}{(1 - \gamma)^2} \tag{12}$$

*where $\pi_w$ is the policy at the $w$-th iteration, $\gamma$ the discount rate, $\delta_{\mathrm{com}}$ the bound on error incurred in computation of policy update and $\epsilon$ the worst-case bound of error on function approximation (Bertsekas & Tsitsiklis, 1997):*

$$\max_{s \in \mathcal{S}} \left| \hat{V}_w(s) - V_{\pi_w}(s) \right| \leq \epsilon, \quad w = 1, 2, \ldots \tag{13}$$

*Proof:* Bertsekas & Tsitsiklis (1997) proves that:

$$\lim_{w \to \infty} \sup \max_{s \in \mathcal{S}} |V_{\pi_w}(s) - V^*(s)| \leq \frac{\delta_{\mathrm{com}} + 2\gamma\epsilon}{(1 - \gamma)^2} \tag{14}$$

where $V^*$ the optimal state value function. Recall that the overall optimization problem is defined as:

$$\max_{\pi \in \Pi} G(\pi) = \sum_{s_0 \in \mathcal{S}} \rho(s_0) V_\pi(s_0) \tag{15}$$

Because $\rho(s_0) \geq 0$ and $\sum_{s_0 \in \mathcal{S}} \rho(s_0) = 1$ for any $s_0 \in \mathcal{S}$, the following equation gives the proof:

$$\begin{aligned}
\lim_{w \to \infty} |G(\pi_w) - G^*| &= \lim_{w \to \infty} \left| \sum_{s_0 \in \mathcal{S}} \rho(s_0) V_{\pi_w}(s_0) - \sum_{s_0 \in \mathcal{S}} \rho(s_0) V^*(s_0) \right| \\
&\leq \lim_{w \to \infty} \sum_{s_0 \in \mathcal{S}} \rho(s_0) |V_{\pi_w}(s_0) - V^*(s_0)| \\
&= \sum_{s_0 \in \mathcal{S}} \rho(s_0) \lim_{w \to \infty} |V_{\pi_w}(s_0) - V^*(s_0)| \leq \frac{\delta_{\mathrm{com}} + 2\gamma\epsilon}{(1 - \gamma)^2}
\end{aligned} \tag{16}$$

$\square$

In practice, $\gamma$ is a part of the environment, and $\delta_{\text{com}}$ can be made small (Bertsekas & Tsitsiklis, 1997). Thus the above theorem shows that decreasing $\epsilon$ is necessary for improving the overall performance. In the following subsection we analyze the relationship between $\epsilon$ and the general design of policy functions.

## 4.2 RISK BOUND

In order to analyze the errors in approximation of policy functions, we transform the loss $\mathcal{L}$ in the optimization target of the policy to a compact and probabilistic form. First we concatenate the denotation of extra information $\kappa_i$ and data sample $(x_j, y_j)$ as $d_{ij} = (\kappa_i, x_j, y_j)$; we consider the number of available $d_{ij}$ to be $n$ and thus use $d_i$ to represent the input of subjectivity. We then use $\boldsymbol{b}$ to denote the weight vector output of the subjectivity $h(d_i)$; then we combine the $N_S$ subjective MDPs base on $\boldsymbol{b}$ as $g(d_i, \boldsymbol{b}) = \boldsymbol{b} \cdot \boldsymbol{g}_S(d_i)$ where $g(\cdot)$ and $\boldsymbol{g}_S$ does not depend on the $\kappa$ part of $d_i$; finally we equally express the original subjectivity in the probabilistic form $h(d_i, \boldsymbol{b}) = P(\boldsymbol{b}|d_i)$ and reach the transformation of $\mathcal{L}$:

$$\mathcal{L}[y_i, \boldsymbol{h}(x_i, \kappa_j) \cdot \boldsymbol{g}_S(x_i)] = \mathbb{E}_{\boldsymbol{b}}\{\mathcal{L}[y_i, g(d_i, \boldsymbol{b})] h(d_i, \boldsymbol{b})\} \tag{17}$$

We denote the distribution of $\boldsymbol{b}$ and $d$ respectively as $F(\boldsymbol{b})$ and $F(d)$; the parameters of $h$ and $g$ are still $\alpha_h$ and $\alpha_S$. Then we define the expected risk and empirical risk as follows:

**Definition 2.** The expected risk functional in subjective reinforcement learning is:

$$Risk(\alpha_h, \alpha_S) = \iint \mathcal{L}[y_i, g(d_i, \boldsymbol{b})] h(d_i, \boldsymbol{b}) \, \mathrm{d}F(\boldsymbol{b}) \, \mathrm{d}F(d) \tag{18}$$

For simplicity, we use $\alpha$ to denote the combined set of $\alpha_h$ and $\alpha_S$, and $\varsigma(d, \boldsymbol{b}, \alpha) = \mathcal{L}[y_i, g(d_i, \boldsymbol{b})] h(d_i, \boldsymbol{b})$. Then the above definition results in:

$$Risk(\alpha) = \iint \varsigma(d, \boldsymbol{b}, \alpha) \, \mathrm{d}F(\boldsymbol{b}) \, \mathrm{d}F(d) \tag{19}$$

We denote the considered number of possible $\boldsymbol{b}$ to be $m$; under specific design of the output of $h(x, \kappa)$, $m$ can be represented using $N_S$, e.g. in one-hot cases we have $m = N_S$. Then we can define the empirical risk functional as follows.

**Definition 3.** The empirical risk functional in subjective reinforcement learning is:

$$Risk_{\text{emp}}(\alpha, m, n) = \frac{1}{m}\sum_{i=1}^{m}\frac{1}{n}\sum_{j=1}^{n} \varsigma(d_j, \boldsymbol{b}_i, \alpha) \tag{20}$$

Note that the $Risk(\alpha)$ here expresses the $\epsilon$ in eq.(12). In order to minimize it, we consider the maximum possible risk gap between $Risk_{\text{emp}}$ and $Risk(\alpha)$:

$$\xi^{\langle m,n\rangle} = \sup_{\alpha}\left[Risk(\alpha) - Risk_{\text{emp}}(\alpha, m, n)\right] \tag{21}$$

The above form of expression enable us to adopt results from Su et al. (2019), which is summarized in the following theorem:

**Theorem 3.** *Consider $\alpha^*$ that minimizes the empirical risk (20), the following inequality takes place with probability $1 - \eta$:*

$$Risk(\alpha^*) < Risk_{\text{emp}}(\alpha^*, m, n) + \zeta_{n,m} \tag{22}$$

*where*

$$\eta = 4\exp\left\{\left[\frac{u_{\boldsymbol{b}}}{m}(1+\ln\frac{2m}{u_{\boldsymbol{b}}}) - \frac{(\zeta_{n,m} - \frac{1}{m})^2}{(B_{\boldsymbol{b}} - A_{\boldsymbol{b}})^2}\right]m\right\} + 4\exp\left\{\left[\frac{\ln m}{n} + \frac{u_d}{n}(1+\ln\frac{2n}{u_d}) - \frac{(\zeta_{n,m} - \frac{1}{n})^2}{(B_d - A_d)^2}\right]n\right\} \tag{23}$$

$A_{\boldsymbol{b}}$, $B_{\boldsymbol{b}}$, $A_d$ and $B_d$ are bounds on functions:

$$A_{\boldsymbol{b}} \leq Risk_{\text{local}}(\alpha, \boldsymbol{b}) \leq B_{\boldsymbol{b}}, \qquad A_d \leq \varsigma(d, \boldsymbol{b}, \alpha) \leq B_d \tag{24}$$

*where* $Risk_{\text{local}}(\alpha, \boldsymbol{b}) = \int \varsigma(d, \boldsymbol{b}, \alpha)\, \mathrm{d}F(d)$; $u_{\boldsymbol{b}}$ *and* $u_d$ *are the VC dimension of function* $h$ *and* $g$ *respectively.*

Note that $A_{\boldsymbol{b}}$, $B_{\boldsymbol{b}}$, $A_d$ and $B_d$ are fixed once the actual form of functions are determined. We also want to emphasize that $\alpha_h^*$ and $\alpha_{\mathrm{S}}^*$, which compose $\alpha^*$, should respectively satisfy eq.(8) and the independence of $g(\cdot)$ on $\kappa$.

## 5 GUIDANCE FOR ALGORITHM DESIGN

In previous section we analyze in theory how the performance of a subjective policy relates to different factors. Now we discuss how these results can be utilized to guide the designing of algorithms for tasks in open environments.

As analyzed in section 3, there is data confusion problem in tasks in open environments. Generally extra information should be introduced, but how to select the capacity of functions so that the overall performance is optimal becomes an important problem, because there are not tasks-specific prior knowledges in open environments. In theorem 2 we derive that, under the assumption of sufficient exploration and sampling that ensures the unbiasedness of data samples, the gap between achieved overall return and the optimal return tends to a value that is bounded according to $\epsilon$; then in theorem 3, this $\epsilon$ is further bounded by $\zeta_{n,m}$ plus the empirical risk with probability $1 - \eta$, considering their maximum values over the whole learning process.

Note that the elements in eq.(23), i.e. $u_{\boldsymbol{b}}$, $u_d$, $m$, $A_{\boldsymbol{b}}$, $B_{\boldsymbol{b}}$, $A_d$ and $B_d$, are all accessible and modifiable as long as the data and the designed functions are determined, and can have concrete meanings if actual algorithms are given. Concretely, $m$ may be the number of options if $\boldsymbol{b}$ is set to be one-hot and the option-based algorithms are adopted; $A_{\boldsymbol{b}}$, $B_{\boldsymbol{b}}$, $A_d$ and $B_d$ may be directly accessible if sigmoid functions are appended to function $h$ and $g$; $u_{\boldsymbol{b}}$, $u_d$ can be determined once the functionals and the mask $\Phi(x, \kappa)$ are given.

Therefore, for tasks in open complex environments, we may first select $\eta$ which reflects the needed confidence level of the analytical results. Then, after data samples from the tasks being collected, eq.(23) guides us to reduce $\zeta_{n,m}$ through the design of elements $u_{\boldsymbol{b}}$, $u_d$, $m$, $A_{\boldsymbol{b}}$, $B_{\boldsymbol{b}}$, $A_d$ and $B_d$, which can then be used for the design of actual forms or hyper-parameters. Note that the expected risk $Risk$ also includes $Risk_{\text{emp}}$ which may also be affected by such adjustments, but as both $Risk_{\text{emp}}$ and $\zeta_{n,m}$ have been expressed explicitly, we argue that a satisfying adjustment is still achievable despite the possible difficulty in computation. may also be affected by such adjustments. Besides, the data samples change as the policy evolves during training, which may lead to the change of ideal adjustments; for this case, changes to $m$, $u_{\boldsymbol{b}}$ and $u_d$ may require the re-tunning of the parameters $\alpha$ in e.g. neural networks; one solution is to calculate adjustments with strict $\eta$ but change them when loose $\eta$ is not satisfied, another solution may require designing a form of parameter-flexible function model.

## 6 CONCLUSION

In this paper we consider tasks in open complex environment and propose data confusion as one core problem. Although current methods relieve this problem to some extent, their reliance on task-specific prior knowledge limits their capacity for tasks in open environments. We propose subjective reinforcement learning framework to represent data confusion problem from a broader view, and subjective policy that generally represents many existing methods. Our theoretical analysis shows the conditions under which a subjective policy outperforms traditional ones, and the relationship between overall performance and the key elements that characterize a subjective policy. As the elements are concrete and modifiable, our analysis further provides guidance for designing algorithms with theoretically guaranteed performance. We also point out several difficulties in practical implementation, which include the complexity in calculating ideal adjustments to the elements and the probable necessity for re-tuning when considering specific kind of models; these problems are planned as future works.

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
