# OpenReview forum: "Subjective Reinforcement Learning for Open Complex Environments"
_ICLR.cc/2020/Conference — Reject_

### Official Review · AnonReviewer1 · 2019-10-18
**Official Blind Review #1**

**Rating:** 3

**Review:**

This paper introduces a new framework for reinforcement learning (named subjective reinforcement learning) which aims to resolve some of the inherent problems with RL in open environments.  The authors posit that one problem with RL in open settings is “data confusion”, which they describe as being situations where there are external factors (e.g. timeframe) that affect the action space differently.  They propose a “subjective reinforcement learning framework” which, as I understand it, can be described as an ensemble of traditional MDP’s subject to external factors k.  The paper evaluates how this subjective policy compares to traditional MDP’s in terms of theoretical bounds on performance.

Although this learning framework seems like a potentially interesting future research direction, I tend to lean towards rejection for the reasons that: (1) the theoretical analysis is difficult to follow and there is sometimes a lack of clarity throughout the paper, (2) it isn’t very clear how easy this framework would be to implement aside from the theoretical guarantees and there aren’t any experiments or proofs of concept that would demonstrate the feasibility or practicality of the proposed framework in a real scenario, (3) the paper would benefit from more discussion of how their work differs from related techniques (like hierarchical RL, various forms of meta-learning, etc.).

Here are some more concrete points:
(1) The feasibility of this framework in a real-world scenario seems a bit hard to imagine and a strong use-case or proof-of-concept would be very helpful.  I liked that this paper provided some theoretical analysis for guiding the design of these systems.  However, it seems like these claims would also benefit from detailed empirical analysis and experiments.  Without empirical results, I feel a bit skeptical about how straightforward it would be to implement such a system or whether it would really be significantly useful in practice.  Similarly, though there are theory-based suggestions for how to optimally design such a system, it might be difficult to implement this system with optimal hyperparameters in a real-world use-case and the challenges in doing so are not really addressed.
(2) In spite of claims that this method is able to be trained without domain knowledge, it seems like domain knowledge would still be necessary for things like determining what external information (K) is available and necessary, determining the appropriate N_S, etc.  It may be helpful for the authors to explain a bit more about how these things can be determined in a truly agnostic way.
(3) It seems like there should be more discussion of the difference from hierarchical reinforcement learning.  In practice, hierarchical RL also can be used in similar ways to what’s described here.  As the authors point out, hierarchical RL is not necessarily splitting into submodels that handle data confusion problems, it seems like that is a constraint that could be added into a hierarchical framework’s design.
(4)  I appreciate that the authors provide detailed theoretical analysis, but it can sometimes be confusing and difficult to follow.  I had some trouble evaluating the correctness of several of the proofs.  It may benefit from re-writing with more concise definitions of all of the variables and more clearly stated assumptions about observable information.  Here are some points of confusion for me:
    - It seems like certain letters (eg. A or K or N_s vs N_d) are being overloaded as variable names with different fonts.  I realize that this is somewhat unavoidable, but I would recommend that the authors try to disentangle the naming a bit for improved clarity.
    - In theorem 1, you stated "the number of all possible data samples tends to infinity when the total number of samples N_d approaches infinity".  This proposition seemed confusingly worded to me.  Maybe I am misunderstanding the wording, but it seems like possibly a tautology?
    - I’m not sure I understand what is meant by “fake subjective policies” in Theorem 1.  Could you explain what is meant by that and the intuition here a bit more?
    - It’s still unclear to me how K (that is, the external information) is being collected at any given timeframe.  Are you assuming that the necessary types of external information corresponding to K (your examples are ‘state history, out-of-MDP task encodings, samples from related tasks, etc.’) have been pre-determined?  If so, how could the most-appropriate type of external information be chosen in a practical way?

I also noticed a few (very minor) grammar errors that authors may want to fix, though they did not affect my review:
- page 1: "tasks in open environments poses difficulties" --> "tasks in open environments pose difficulties"
- page 1: "Problem is that both..." --> "The problem is that both..."
- page 2: "we propose a novel framework named as Subjective reinforcement" --> "we propose a novel framework named Subjective reinforcement"
- page 4: "should contain no data confusion" --> "should not contain data confusions"

**Experience Assessment:**

I do not know much about this area.

**Review Assessment: Checking Correctness Of Derivations And Theory:**

I assessed the sensibility of the derivations and theory.

**Review Assessment: Checking Correctness Of Experiments:**

N/A

**Review Assessment: Thoroughness In Paper Reading:**

I read the paper thoroughly.

---

> ### Author Response · Authors · 2019-11-12
> **Author response to Review #1**
>
> We appreciate the time you took reviewing our submission and hope our response help address some of your concerns.
>
> Replies to the main part of review:
> (1) Q: “the theoretical analysis is difficult to follow and there is sometimes a lack of clarity throughout the paper”
> A: We noticed some mistakes in symbols and definitions of variables and corrected them in the updated version. Hope the new version can make it easier to understand.
>
> (2) Q: “it isn’t very clear how easy this framework would be to implement aside from the theoretical guarantees and there aren’t any experiments or proofs of concept that would demonstrate the feasibility or practicality of the proposed framework in a real scenario”
> A: As we mentioned in section 5, we realize that there may be some serious difficulties in application of our framework to real scenes. In fact, the main intent of this paper is to take a step forward in theory towards algorithm design considering general cases of reinforcement learning tasks. We acknowledge that concrete examples may well help exhibit the problem we are considering and prove the practicability of our work.
>
> (3) Q: “the paper would benefit from more discussion of how their work differs from related techniques (like hierarchical RL, various forms of meta-learning, etc.).”
> A: In section 2 we briefly introduce some closely related fields. We summarize that techniques in these fields are introducing task-specific designs to reach better performance in certain kinds of tasks. Our work hope to unify such techniques and provide efficient algorithm designs without human prior. From this perspective, we agree that more discussions can help strengthen our conclusion and clarify our idea, but we think there is no other comparable differences between our work and other concrete techniques. Please see (3) in the part below for one example.
>
> Replies to the concrete points:
> (1) Please see replies to question 2 in above part.
>
> (2) In our framework, “\kappa” denotes the information provided by the environment other than current state “s_t”, and so our method only chooses how to utilize it according to its actual instantiation in tasks. For a concrete example, consider our subjective “\bold{h}” is designed to be a one-hot vector, then we get m=N_S; the actual data (including \kappa) and the chosen form of loss function “\script{L}” give concrete bounds defined in eq. (24); given expected \zeta and \eta, inequation (23) gives us the relationships m, u_b, u_d (VC dimensions of function approximators) should obey. In this way, we can determine N_S with given data and without domain knowledge.
>
> (3) In hierarchical RL the hyperparameters of function approximators (i.e. the layer num of neural networks) and the maximum number of lower-level policies are designed by designer; while in our work we propose to adjust them according to actual data automatically and with theoretical guarantee, based on our analysis of the relationship between overall performance and some key variables.
>
> (4)
> a) Thanks for your advice. We will pay more attention to the simplicity of notations in our future work.
> b) We wanted to express “the number of each kind of data samples tends to infinity” as the condition for theorem 1, to remove the possibility of insufficient exploration. To make it clearer, we update it to “all possible data samples appear infinite times”.
> c) The main idea in the proof of theorem 1 is to show that any policy in original forms can be expressed by a subjective policy. We use “fake” to emphasize that the defined “\pi_{z,fake}” shares the same form as subjective policy but is actually equal to the policy in the original form.
> d) Yes, we are assuming \kappa is pre-determined with the tasks. In this paper we only focus on how to use given information with theoretical guarantee. When there are more than one sources (types) of \kappa available, we may consider applying eq. (23) to these sources respectively and select one according to some preferences, i.e. affordable VC dimension of function approximators.

---

### Official Review · AnonReviewer2 · 2019-10-23
**Official Blind Review #2**

**Rating:** 3

**Review:**

The paper introduces the Subjective Reinforcement Learning framework to formalize the problem of using extra information to split large, nonstationary environments into separate, simple, stationary MDPs. First, the paper introduces and motivates the problem on a high level: the same state-action pairs may transit to different successive states and rewards because of nonstationary dynamics/reward function, or variance in the environment or tasks. This phenomenon is termed "data confusion". The paper then summarizes some related approaches to dealing with this phenomenon. Next, the paper introduces the subjective RL framework in detail in section 3:
- Extra information (kappa) is needed to resolve the data confusion.
- The "subjectivity" (h) is a function that maps the extra information to a vector of weights over "subjective" MDPs.
- A policy is maintained for each subjective MDP, and the overall policy is the vector product of h and the vector of subjective policies.
In section 4, the paper presents 3 theorems arguing that using the subjective RL framework doesn't harm performance. The paper then gives brief guidelines for designing algorithms using the subjective RL framework before concluding.

At the present time I recommend rejecting the paper. It does not actually present a concrete solution method, instead simply giving brief guidelines for the reader to design algorithms by. The subjective RL framework unifies and subsumes several existing approaches, but I don't feel that in itself is a significant enough contribution to warrant publication. The theorems presented essentially argue that using the subjective RL framework does not harm performance, but there are no mentions of the computational costs involved with maintaining policies for each subjective MDP. In addition, it's not clear where the subjective MDPs come from.
The paper also had issues with clarity, including many grammatical errors.

I think this paper tackles an important problem from an interesting point of view, but stops short of giving a concrete algorithm that can be implemented and tested. It seems like a good candidate for a workshop, which could be a good opportunity for discussion and feedback.

General comments:
- The definition of rewards is confusing; script R was never defined.
- "minimize objective 1" should be actually be "maximize objective 1"?
- Rewards are usually defined over state-action pairs, not just states. Why choose this unconventional formulation for rewards?

**Experience Assessment:**

I do not know much about this area.

**Review Assessment: Checking Correctness Of Derivations And Theory:**

I assessed the sensibility of the derivations and theory.

**Review Assessment: Checking Correctness Of Experiments:**

N/A

**Review Assessment: Thoroughness In Paper Reading:**

I read the paper at least twice and used my best judgement in assessing the paper.

---

> ### Author Response · Authors · 2019-11-12
> **Author response to Review #2**
>
> Thank you for taking the time to review our submission and for your feedback. We hope the following may address some of your concerns.
>
> Replies to the main part of review:
> We regret not having made it clearer about our design of the theorems:
> 1) Theorem 1 aims to provide a basic result that the subjective policy will not get worse results than the original form of policy; this is quite easy to get and not practically useful.
> 2) In theorem 2 we continue to analyze the relation between the bound of performance of a converged policy (lim|G(\pi)-G*|) and the worst-case bound of error on function approximation (\epsilon); this enables us to analyze the final performance through analyzing the function approximator, where the subjective policy makes a difference.
> 3) Then in theorem 3 we get the relationship between the risk bound of the function approximator and some variables that are determined by data and the hyperparameters of selected function model; this enables us to control the overall performance through adjusting the related variables according to the data got in specific tasks.
>
> In this paper, we wish to take a step forward in theory towards algorithm design with considering general cases of reinforcement learning tasks. As we analyzed in section 5, currently the variables (e.g. m, u_b, u_d) can theoretically be adjusted but there do exist difficulties when considering some function models, and we acknowledge that concrete examples may well help prove the practicability of our work.
>
> Replies to “general comments”:
> 1) Q: “The definition of rewards is confusing; script R was never defined.”
> A: We noticed this mistake and have corrected it in the updated version.
>
> 2) Q: "minimize objective 1" should be actually be "maximize objective 1"?
> A: This is another mistake in writing. Thanks for your correction. Again, we corrected it in the updated version.
>
> 3) Q: Rewards are usually defined over state-action pairs, not just states. Why choose this unconventional formulation for rewards?
> A: We recognize that in many publications the reward function is defined as R_0(s, a). In this paper, we are not introducing anything special by defining reward function as R(s). In fact, we think these forms are equal, because R_0(s, a)=E_{s’}[R(s’)P(s’| s, a)] (here s’ stands for the next state).

---

> > ### Comment · AnonReviewer2 · 2019-11-14
> > **Reviewer reply to the author response**
> >
> > Thank you for reading my review and addressing some of my comments.
> >
> > The authors' summary of the motivation for the proofs is helpful for understanding why they were included in the paper. However, I'm still concerned about the computational cost of the proposed framework (i.e., maintaining value functions and/or policies for each subjective MDP), and would still have liked to have seen an implementable algorithm with an experiment showing it can work.

---

### Official Review · AnonReviewer3 · 2019-10-24
**Official Blind Review #3**

**Rating:** 1

**Review:**

The paper suggests that one common problem encountered by reinforcement learning algorithms in open environments is "data confusion", which essentially means showing the same input data with different --possibly contradictory-- labels/targets.

The proposed solution to this conceptual problem is to split the original MDP "M" up into multiple simpler MDPs "Mk", where M does contain possibly contradictory ("confusing") data, while each individual Mk does not contain any such problem and, even better, is stationary.

The "subjectivity" function "h" then has as role to split any data tuple across Mk, possibly using extra information kappa.

Furthermore, several theorems show that under several conditions, the return of the subjective policy (learned via Mk) is not worse than that of the "traditional" policy.


I lean towards rejecting this paper. The whole gist of the framework can be crudely summarized as "if data contradicts, split up into non-contradictory sets using extra info." The motivation keeps repeating that no task-specific prior knowledge being necessary, but I believe this hinges on "h" being sensible, which might not be feasible without task-specific prior knowledge.

Furthermore, and this is my main concern, there is not a single experiment demonstrating how any of this would behave in practice. It would be good to have one (possibly constructed) experiment showing that data confusion indeed is a problem in practice (intuitively, it is), and then a specific instantiation of the framework that solves this example. Furthermore, I am not convinced that the proposed bounds can easily be concretized for an instantiation of the proposed framework, especially when considering deep networks; again, this concern could be alleviated by an example instantiation. Proposing something that is in principle more general and "in principle cannot be worse" but then not demonstrating that it actually is the case is, in my opinion, not enough.



Finally, and this is not a deciding factor in my rating, the paper has quite some writing problems. On the first page alone, I found a lot of spelling and grammatical mistakes (see list at end) and the notation is sometimes confusing to me. For example, "R" is defined as a mapping of S x /R -> [0,1], but what is "/R" (curly R)? And then in (1) R is used with a single argument while in (2) not anymore. I can guess what is meant, but it feels inconsistent. In Theorem 1, I believe it should be "the gap \delta >= 0" and not "the gap g >= 0", no?

Abstract and 1st paragraph mistakes (unfortunately, no line numbers in this template!): "researches" -> "research", "algorithm designing" -> "algorithm design", "task-specific prior knowledge about tasks." -> "task-specific prior knowledge.", "not known in prior" -> "not known a priori", "Classical RL model environment..." -> "Classical RL models environment...".
Also, quite some citations are missing the year, e.g. Schaul et al., Papavassiliou&Russell, ...

**Experience Assessment:**

I have published one or two papers in this area.

**Review Assessment: Checking Correctness Of Derivations And Theory:**

I assessed the sensibility of the derivations and theory.

**Review Assessment: Checking Correctness Of Experiments:**

N/A

**Review Assessment: Thoroughness In Paper Reading:**

I read the paper at least twice and used my best judgement in assessing the paper.

---

> ### Author Response · Authors · 2019-11-12
> **Author response to Review #3**
>
> Thank you for your detailed and insightful review. We hope the following address some of your concerns.
>
> Q: “I believe this hinges on "h" being sensible, which might not be feasible without task-specific prior knowledge.”
> A: In this paper, “h” is used as one component to model the decomposition of the policy according to “\kappa” i.e. information provided by the environment other than current state “s_t”. We acknowledge that 1) in cases where there are not enough “\kappa” to decompose the policy and thus to avoid data confusion, our proposed framework cannot bring improvements, and 2) when there is task-specific knowledge available, “h” can be designed in a task-specific way and the overall performance may be better. However, we think these does not mean “h” relies on task-specific prior knowledge. Despite the difficulties in application, the proposed guidance for algorithm bases on eq. (23), which consists of only properties of the data collected, but not prior knowledge. Maybe we should have made it clearer that, the intent of this paper is to take a step forward in theory towards efficient algorithm design considering general cases of reinforcement learning tasks.
>
> Q: “there is not a single experiment demonstrating how any of this would behave in practice”; “I am not convinced that the proposed bounds can easily be concretized for an instantiation of the proposed framework, especially when considering deep networks”
> A: We acknowledge that the lack of example instantiations makes our work not satisfactorily convincing, but we tend to take a step forward in theory and focus on more general cases. Concrete examples and algorithm instantiations are planned as future work.
> As for concerns about neural networks, our result eq. (23) requires “u_b” and “u_d” i.e. the VC dimension of functions, which have been analyzed by some works such as [1]. If we construct useful relationships between hyperparameters of neural networks and its VC dimension, our theoretical result can be applied.
>
> Q: “what is "/R" (curly R)?”
> A: We are sorry for the mistake in some symbols. In previous version we wanted to use “normal R” to denote the reward function defined on the state space, and the “curly R” to denote the set of possible reward values. In the updated version we correct this mistake, i.e. only “normal R” is used and is in the form of a probability distribution. Further, we expand the notation for risk functions to “Risk(...)” to avoid confusion.
>
> Q: “In Theorem 1, I believe it should be "the gap \delta >= 0" and not "the gap g >= 0", no?”
> A: Thanks for your correction. We fixed this mistake in the updated version.
>
> Q: some writing problems
> A: In our updated version, we have corrected some mistakes in grammar and citation, including the ones mentioned in this problem.
>
> [1] Sontag, E. D. (1998). VC dimension of neural networks. NATO ASI Series F Computer and Systems Sciences, 168, 69-96.

---

> > ### Comment · AnonReviewer3 · 2019-11-15
> > **Thank you for the response.**
> >
> > Roughly speaking, the updated paper changes some writing problems, replaces sums by integrals around R, and replaces "R" by "Risk" where appropriate. While it is a small improvement, the whole paper still lacks a lot of clarity - a sentiment reflected by both other reviewers too.
> >
> > This also does not address any of the concerns. I am still of the opinion that the framework automatically segregating the MDP into individual, simpler MDPs based on no prior knowledge whatsoever is an extraordinary claim, requiring extraordinary evidence. If this was supported by a concrete instantiation of the framework and effectiveness of it demonstrated on at least some tasks, my rating would be very different. As-is, I am not ready to recommend acceptance of this work.

---

### Decision · Program_Chairs · 2019-12-19

**Decision:**

Reject

**Comment:**

The authors propose a learning framework to reframe non-stationary MDPs as smaller stationary MDPs, thus hopefully addressing problems with contradictory or continually changing environments. A policy is learned for each sub-MDP, and the authors present theoretical guarantees that the reframing does not inhibit agent performance.

The reviewers discussed the paper and the authors' rebuttal. They were mainly concerned that the submission offered no practical implementation or demonstration of feasibility, and secondarily concerned that the paper was unclearly written and motivated. The authors' rebuttal did not resolve these issues.

My recommendation is to reject the submission and encourage the authors to develop an empirical validation of their method before resubmitting.